# Simultaneous Quantitative Determination of Residues of Pyriproxyfen and Its Metabolites in Tea and Tea Infusion Using Ultra-Performance Liquid Chromatography–Tandem Mass Spectrometry

Hongxia Li [1,2], Qing Zhong [1,2], Xinru Wang [1], Fengjian Luo [1], Li Zhou [1], Zongmao Chen [1] and Xinzhong Zhang [1,*]

[1] Tea Research Institute, Chinese Academy of Agricultural Sciences, Hangzhou 310008, China; lhx_tricaas@163.com (H.L.); zhongqing@yili-tea.com (Q.Z.); wangxinru@tricaas.com (X.W.); lfj@tricaas.com (F.L.); lizhou@tricaas.com (L.Z.); zmchen2006@163.com (Z.C.)
[2] Graduate School of Chinese Academy of Agricultural Sciences, Beijing 100081, China
* Correspondence: zhangxinzhong@tricaas.com or zxz.1982@163.com; Tel.: +86-571-87963516; Fax: +86-571-86650331

**Abstract:** A reliable, simultaneous residue-analysis method for pyriproxyfen and its five metabolites in fresh tea leaves, green tea, black tea, green-tea infusion and black-tea infusion was developed and validated. The samples were extracted with acetonitrile, purified using a modified QuEChERs (quick, easy, cheap, effective, rugged and safe) method and determined using ultra-performance liquid chromatography–tandem mass spectrometry (UPLC-MS/MS). The method was successfully calibrated in the range of 0.005–2.50 mg/L with correlation coefficients ($r$) equal to or above 0.9957. The limits of detection (LODs) were less than 0.002 mg/L. The average spiked recoveries of pyriproxyfen and its metabolites at four levels were 71.2~102.9% with relative standard deviations (RSDs) of 0.3~14.4%. The limits of quantification (LOQs) in fresh tea leaves, tea and tea infusion were 0.002 mg/kg, 0.005 mg/kg and 0.0002 mg/L, respectively. This proposed method was feasible and was applied to analyze the residues of pyriproxyfen and its five metabolites on real fresh tea leaf samples. The results indicated that the half-life ($t_{1/2}$) of pyriproxyfen on fresh tea leaves was 2.48 d, and the five metabolites were detected on fresh tea leaves during field growth after application.

**Keywords:** tea; pyriproxyfen; metabolite; residue analysis; UPLC-MS/MS

## 1. Introduction

Tea (*Camellia sinensis L.*) is an important economic crop and one of the most popular drinks in the world. With the continuous expansion of the tea industry in China, pesticides are commonly used to improve tea yield and quality, which has led to the large and widespread use of chemical pesticides in tea farming. The registered pesticides in tea farming are few, and their long-term, single and frequent usage results in pesticide resistance and low-control effects. As a result, large quantities of pesticides are used, and that leads to high residues in tea. At present, the maximum residue limits (MRLs) of pesticides in tea have been changed in China and abroad. In recent years, neonicotinoids have been restricted in tea plantation due to their high leaching rates in tea brewing. Therefore, it is extremely urgent to find new alternative pesticides to prevent and control diseases and insect pests in tea farming. Before that, it is necessary to establish a residue-analysis method for pesticides, including their metabolites, to clarify their residue behavior and safety.

Pyriproxyfen is a broad-spectrum insect-growth regulator that interferes with insect development via a distinctive mechanism action, and it was first synthesized and developed by Sumitomo Chemical Co., Ltd. (Tokyo, Japan) [1,2]. Pyriproxyfen is more selective and consequently safer for non-target organisms than conventional insecticides [3] and exerts high biological activity against public-health insect pests [4–6] and agricultural and

horticultural pests [7–9]. Meanwhile, pyriproxyfen is also recommended for addition at the concentration of 0.01 mg/L to drinking water for public-health and vector-control programs by the World Health Organization (WHO) [10]. It can effectively inhibit a large number of agricultural pests for increasing tea production, which has a great potential application in tea gardens [11], and also has a certain control effect on Tea Lace Bug (*Stephanitis chinensis* Drake) [12]. However, pyriproxyfen does not cause death in rats [13], teratogenicity in rats and rabbits [14], central-nervous-system development issues in zebrafish [15] nor meaningful mortality or abnormal behavior in tadpoles [16]. The toxicities of some metabolites [14] in different environment matrices are even greater than that of pyriproxyfen itself [17]. Especially for its low solubility (0.367 mg/L, 25 °C) and high hydrophobicity (its octanol water partition coefficient is log *Kow* = 5.37, 25 °C), pyriproxyfen easily accumulates in animal fat [18], so the potential risks of the residues of pyriproxyfen and its metabolites cannot be ignored.

Several chromatographic analytical methods combined with various spectroscopic techniques for pyriproxyfen residues on different vegetable and fruit samples [19,20], such as fresh and canned peach [21], grapes [22], peppers [23,24], mushrooms [25], tomatoes [23,26,27] and citrus [28], have been reported in previous studies. There have also been a few studies focused on the residue degradation of pyriproxyfen and its metabolites in rats and mice [29], microsomes of housefly larvae [30], tadpoles of African clawed frogs [16], rat liver [17], water-sediment system [31] and soil [32]. These studies commonly use QuEChERS (quick, easy, cheap, effective, rugged and safe) methods [33] or solid-phase extraction (SPE) combined with gas or liquid chromatography (GC or LC) coupled to different detectors, such as nitrogen–phosphorus detectors (NPDs), ultraviolet detectors (UV), diode-array detectors (DADs), mass spectrometers (quadrupole time-of-flight (QqTOF), or triple quadrupole (QqQ) or quadrupole ion trap (QIT)), etc.

Tea is richer in polyphenols, caffeine, pigments and other components than other agro-products, and there are also differences in the contents of different types of tea. Moreover, if the interferents cannot be effectively purified, they affect the analysis of target compounds [34] and produce different matrix effects [35]. So, these existing established methods are not suitable for the analysis of pyriproxyfen residues in different tea matrices. All of these analytical methods were only applied to pyriproxyfen, and there are still scarce studies on the analysis of its metabolites in different samples. Only Wu et al. reported on the residual analysis of pyriproxyfen and its four metabolites (4′-OH-Pyr, 2-OH-PY, DPH-Pyr and 4″-OH-POP) in bee products [36]. There is no analytical method for the determination of neither pyriproxyfen residues in tea and tea infusion nor its metabolites.

Therefore, this study aims to establish a simultaneous residue-analysis method for pyriproxyfen and its five corresponding metabolites (PYPA, PYPAC, DPH-Pyr, 5″-OH-Pyr and 4′-OH-Pyr) in fresh tea leaves, tea and tea infusion, which is of great significance for providing a qualitative and quantitative analysis method for subsequent research on the residue-degradation behavior and dietary risks of pyriproxyfen and its metabolites in the tea growing–processing–brewing chain. It is our further aim to furnish basic scientific data to guide the rational usage of pyriproxyfen in tea farming, to formulate the relevant maximum residue limit (MRL) standard for tea and to conduct the risk assessment of human tea drinking.

## 2. Materials and Methods

### 2.1. Materials and Reagents

Pyriproxyfen ($C_{20}H_{19}NO_3$; CAS No. 95737-68-1; 98.0% purity) was purchased from Dr. Ehrenstorfer GmBH (Augsburg, Germany). 4′-OH-Pyr ($C_{20}H_{19}NO_4$; CAS No. 159600-61-0; 95.6% purity), PYPA ($C_8H_{11}NO_2$; CAS No. 133457-51-9; 95.3% purity), PYPAC ($C_8H_9NO_3$; CAS No. 168844-45-9; 99.6% purity), DPH-Pyr ($C_{14}H_{15}NO_3$; CAS No. 142346-93-8; 95.1% purity) and 5″-OH-Pyr ($C_{20}H_{19}NO_4$; CAS No. 168844-46-0; 95.1% purity) were commissioned to be synthesized in a laboratory. The chemical structures of pyriproxyfen and the five metabolites are shown in Figure 1, and the LC-MS and NMR information of the five

metabolites are shown in Figure S1 and Table S1. We obtained 100 g/L of pyriproxyfen emulsifiable concentrate (EC) from Shanghai Shengnong Biochemical Products Co., Ltd. (Shanghai, China). Acetonitrile (MeCN), methanol (MeOH), acetic acid (HAc) and formic acid (FA) were of HPLC grade; the first two were purchased from Merck (Darmstadt, Germany), and the latter two were purchased from CNW Technologies GmbH (≥98.0%; Düsseldorf, Germany). Sodium acetate (NaAc), sodium chloride (NaCl), anhydrous magnesium sulfate (MgSO$_4$), NH$_3$·H$_2$O and ammonium acetate (AMA) were of analytical grade and were purchased from Shanghai No.4 Reagent & H.V Chemical Co., Ltd. (Shanghai, China). Purified water was purchased from Hangzhou Wahaha Group Co., Ltd. (Hangzhou, China). Regarding the absorbents, bamboo charcoal (BC; 100 nm) was purchased from Shanghai Hainuo Carbon Industry Co., Ltd. (Shanghai, China), and octadecyl silane (C$_{18}$; 50 μm), polymeric weak anion exchange (PWAX; 40–60 μm), graphitized carbon black (GCB; 120–140 mesh), primary secondary amine (PSA; 40–60 μm), octadecyl silane-N (C$_{18}$-N; 40–60 μm), strong cation exchange (SCX; 40–60 μm) and 0.22 μm filter film were purchased from Bonna Agela Technologies Co., Ltd. (Tianjin, China).

**Figure 1.** The chemical structures of pyriproxyfen and its metabolites PYPA, PYPAC, DPH-Pyr, 5″-OH-Pyr and 4′-OH-Pyr.

Standard stock of pyriproxyfen and its five metabolites at a concentration of 200 mg/L was obtained by accurately weighing quantities of 0.0100 g of pyriproxyfen and its five metabolite standards and dissolving them in 50 mL volumetric flasks with MeCN, respectively. Then, they were stored at −20 °C until use.

Blank fresh tea leaves were obtained from a well-growing tea plantation without pesticides at Shengzhou Experimental Base, Tea Research Institute, Chinese Academy of Agricultural Sciences (29.7421° N, 120.8189° E; located in Shengzhou City, Zhejiang, China) to ensure tea was without residues of pyriproxyfen or its metabolites, to then process the blank fresh tea leaves into blank green tea and black tea and to obtain blank green-tea infusion and black-tea infusion according to the relevant process steps.

### 2.2. UPLC-MS/MS Analysis Method

An UPLC-MS/MS system that consisted of Waters Acquity UPLC® H-Class Ultra Performance Liquid Chromatography (including Quaternary Solvent Manager, FTN Sample Manager and Column Manager) and Waters Xevo® TQ-S Micro Triple Quadrupole Mass Spectrometer equipped with an electrospray ionization (ESI) source (Waters Corp., Milford, MA, USA) was used for the analyses of pyriproxyfen and its metabolites. Instrument control, data acquisition and data analyses were performed using the MassLynx 4.1 workstation.

An Acquity UPLC HSS T3 column (100 mm × 2.1 mm, 1.8 μm; Waters Corp., Milford, MA, USA) was used for the separation of pyriproxyfen and its five metabolites. The column temperature was maintained at 40 °C. The mobile-phase systems were MeOH with 0.1% FA and $H_2O$ with 0.1% FA. The flow rate was 0.25 mL/min, and the injection volume was 5 μL.

Electrospray positive ionization (ESI+) in multiple reaction monitoring (MRM) mode was used for the UPLC-MS/MS analyses of pyriproxyfen and its five metabolites. The source temperature was 150 °C. The capillary electrospray voltage was 3.5 kV. The desolvation temperature was 350 °C. The desolvation gas and cone gas were nitrogen (99.95%) at flow rates of 750 L/h and 50 L/h, respectively. The collision gas was argon (99.995%). The other optimal parameters of the UPLC-MS/MS analysis method are listed in Table 1.

**Table 1.** The optimal UPLC-MS/MS analysis method for the detection of pyriproxyfen and its metabolites.

| Time (min) | Flow Rate (mL/min) | Phase A MeOH (0.1% FA) (%) | Phase B $H_2O$ (0.1% FA) (%) | Compounds | Retention Time (min) | Parent Ions (*m/z*) | Daughter Ions (*m/z*) | Cone Voltage (V) | Collision Energy (V) |
|---|---|---|---|---|---|---|---|---|---|
| 0 | 0.25 | 30 | 70 | PYPA | 2.71 | 154.0 | 95.9 * | 15 | 10 |
| 2.0 | 0.25 | 70 | 30 | | | | 77.9 | | 25 |
| 6.5 | 0.25 | 95 | 5 | PYPAC | 3.41 | 168.1 | 95.9 * | 20 | 15 |
| | | | | | | | 121.9 | | 10 |
| 8.0 | 0.25 | 100 | 0 | DPH-Pyr | 4.65 | 246.0 | 95.9 * | 15 | 10 |
| | | | | | | | 150.9 | | 10 |
| 9.8 | 0.25 | 100 | 0 | 4′-OH-Pyr | 6.15 | 338.0 | 95.9 * | 18 | 15 |
| | | | | | | | 243.0 | | 15 |
| 10.3 | 0.25 | 30 | 70 | 5″-OH-Pyr | 6.72 | 338.0 | 111.9 * | 22 | 15 |
| | | | | | | | 227.0 | | 15 |
| 12.0 | 0.25 | 30 | 70 | Pyriproxyfen | 7.63 | 322.0 | 95.9 * | 20 | 15 |
| | | | | | | | 227.0 | | 15 |

* represents quantitative ion pairs.

### 2.3. Sample Preparation, Extraction and Purification

2.3.1. Fresh Tea Leaves, Green Tea and Black Tea

We accurately weighed 2.00 ± 0.01 g of green tea or black tea (or 5.00 ± 0.01 g of fresh tea leaves) into a 50 mL centrifuge tube, and 10.0 mL (or 5.0 mL for fresh tea leaves) of $H_2O$ with 5% FA was added for soaking and vortexed for 30 min. Afterwards, 10.0 mL of MeCN with 2% FA was added into the tube, extracted and vortexed for 2 h, followed by homogenization for 1–3 min at 17,400 r/min. Then, 3.0 g NaCl, 0.5 g NaAc and 5.0 g $MgSO_4$ (or 3.0 g for fresh tea leaves) were added into the tube, which was shaken for 5 min and sonicated for 10 min. After being centrifuged for 5 min at 7500 r/min, the whole supernatant was transferred into a 250 mL round-bottomed flask, and the lower solid was extracted again with 10.0 mL of MeCN with 2% FA, combined with the supernatant into the flask and concentrated to dryness using rotary evaporation at 42 °C. Then, 10.0 mL of MeCN with 2% FA was added, after being ultrasonically re-dissolved, and the solution was transferred into another 10 mL tube and centrifuged for 5 min at 4000 r/min. Afterwards, 1.5 mL of the supernatant was placed into a 2 mL centrifuge tube that contained 10 mg of GCB, 50 mg of $C_{18}$, 50 mg of $MgSO_4$ and 10 mg of PWAX. The tube was shaken for 5 min and centrifuged for 5 min at 12000 r/min; then, the contents were filtered through 0.22 μm

filter film into an injection bottle, analyzed using UPLC-MS/MS and quantified using the matrix-matched external standard calibration method.

### 2.3.2. Green-Tea Infusion and Black-Tea Infusion

According to the tea/water ratio of 1/50 (3.00 g of tea was infused in 150 mL of boiling water), green tea or black tea was brewed for 10 min in boiling water (100 °C) and filtered to obtain the corresponding tea infusion. Then, 20.0 mL of tea infusion was placed into a 50 mL centrifuge tube, and 20.0 mL of MeCN with 2% FA was added; then, the solution was vortexed. Quantities of 1.0 g of NaAc and 8.0 g of NaCl were added into the tea infusion, which was mixed and shaken for 5 min, followed by centrifugation for 5 min at 7500 r/min. Then, the whole supernatant was transferred into a 150 mL round-bottomed flask. After that, 15.0 mL of MeCN with 2% FA was added into the tube for the second extraction; the two supernatants were combined into the 150 mL round-bottomed flask and concentrated to dryness using rotary evaporation at 42 °C. Then, the residues in the flask were re-dissolved with 1.0 mL of MeCN, filtered through 0.22 µm filter film into an injection bottle, analyzed using UPLC-MS/MS and quantified using the matrix-matched external standard calibration method.

### 2.4. Method Validation

The sensitivities, recoveries, precisions and specificities of pyriproxyfen and its five metabolites in fresh tea leaves, tea and tea infusion obtained using this residue-analysis method were calculated.

The linearities were calculated both in solvents and different tea matrices using matrix-matched calibration curves prepared as described above in Section 2.3 at the concentration range of 0.005~2.50 mg/L. Each concentration was determined three times. The limits of determination (LODs) were defined by the responses to the lowest concentration level on the matrix standard curve when signal-to-noise ratio $S/N = 3$.

The matrix effects (MEs) were evaluated by comparing the responses to the different matrix standard solutions ($S_m$) with the response to the standard solution ($S_s$) according to equation ME = $(S_m - S_s)/S_s$ *100%, where ME > 0 indicates a matrix-enhancing effect, while ME < 0 indicates a matrix-suppressing effect [37,38].

The recovery studies of pyriproxyfen and its five metabolites in different blank matrices were performed using the matrix external standard method and carried out at four spiked levels with five replicates at each level for fresh tea leaves (0.002, 0.01, 0.10 and 1.00 mg/kg), tea (0.005, 0.025, 0.25 and 2.50 mg/kg) and tea infusion (0.0002, 0.001, 0.01 and 0.10 mg/L), respectively. The limits of quantification (LOQs) were defined as the lowest spiked levels with suitable recoveries of 70~120% and relative standard deviations (RSDs) lower than 20% [39].

### 2.5. Field Test and Analysis of Real Fresh Tea Leaf Samples

Field trials of 100 g/L of pyriproxyfen EC on tea plants (Longjing 43) were conducted at Shengzhou Experimental Base, Tea Research Institute, Chinese Academy of Agricultural Sciences (29.7421° N, 120.8189° E; Zhejiang, China) from July to August 2019. The soil type, pH, organic matter content and cation-exchange capacity (CEC) were reddish-brown clay soil, 4.20, 4.55% and 21.5 cmol (+)/kg, respectively. The actual temperatures during the field trial were 20 °C~37 °C.

The experimental treatment was performed according to Guidelines on Pesticide Residue Trials in China (NY/T 788-2018) and consisted of three replicate plots with an area of 22 m² and a control plot without pyriproxyfen. Then, 100 g/L of pyriproxyfen EC dissolved in water was applied to the three tea plots at 7.5 g a.i./100 m². After application, at different time intervals (0 (2 h), 1, 2, 3, 5, 7, 10, 14 and 21 days), 2 kg of representative fresh tea leaf samples (one bud and 2 or 3 leaves) was randomly collected from at least 15 sampling points; then, it was crushed and stored at −18 °C prior to analysis.

### 3. Results and Discussion

*3.1. Optimization of UPLC-MS/MS Conditions*

3.1.1. Optimization of Chromatographic Conditions

Previous studies showed that the mobile phases used for pyriproxyfen were MeOH and $H_2O$ [28,40,41], MeCN and $H_2O$ [31,42,43], 0.01% trifluoroacetic acid and MeCN [44], and 0.03% HAc in MeCN and 0.03% HAc in $H_2O$ [16]. In this study, the combinations of 0.1% FA in MeCN–10 mmol/L AMA in $H_2O$, 0.1% FA in MeOH–10 mmol/L AMA in $H_2O$, 0.1% FA in MeCN–0.1% FA in $H_2O$ and 0.1% FA in MeOH–0.1% FA in $H_2O$ were used as the mobile phases, respectively, and the separation effect of pyriproxyfen and its metabolites were compared. The results indicated that when the combination of 0.1% FA in MeOH–0.1% FA in $H_2O$ was chosen as the mobile phase, pyriproxyfen and its metabolites achieved the ideal separation effect. The optimized mobile-phase gradient programs are listed in Table 1, and the chromatograms of pyriproxyfen and its five metabolites are shown in Figure S2.

The injection solvent has an effect on the peak shapes and responses of analytes [45]. The differences in the peak shape and peak intensity of pyriproxyfen and its five metabolites at different ratios (10/0, 9/1, 8/2, 7/3, 6/4, 5/5, 4/6, 3/7, 2/8 and 1/9, *v/v*) of MeCN/$H_2O$ were compared in this study. The results showed that different ratios of MeCN/$H_2O$ had little effect on the peak shape of pyriproxyfen and its five metabolites, while they had a large effect on the peak intensity (Figure S3). As the proportion of $H_2O$ in the injection solvent increased, the peak intensity of pyriproxyfen and 4′-OH-Pyr decreased; that of 5″-OH-Pyr decreased and then increased; that of DPH-Pyr greatly fluctuated; that of PYPAC increased and then decreased; and that of PYPA gradually increased. Therefore, MeCN/$H_2O$ at 10/0 was chosen, that is, MeCN was chosen as the injection solvent.

3.1.2. Optimization of MS/MS Conditions

Electrospray ionization at full scan in positive mode (ESI$^+$) and negative mode (ESI$^-$) was conducted to analyze pyriproxyfen and its metabolites (PYPA, PYPAC, DPH-Pyr, 5″-OH-Pyr and 4′-OH-Pyr) to obtain the best ionization mode. Pyriproxyfen and its five metabolites had the best ionization effects and the highest responses when subjected to the ESI$^+$ mode with [M+H]$^+$ at *m/z* 322.0, 154.1, 168.1, 246.0, 338.0 and 338.0, respectively.

Different cone voltages (5 V–50 V) and different collision energies (5 V–50 V) were optimized to obtain the maximum sensitivity and minimum interference of the residual analyses of pyriproxyfen and its five metabolites. When the cone voltages were 20, 15, 20, 15 and 22 V, the quasi-molecular ions of pyriproxyfen (*m/z* 322.0), PYPA (*m/z* 154.1), PYPAC (*m/z* 168.1), DPH-Pyr (*m/z* 246.0), 5″-OH-Pyr (*m/z* 338.0) and 4′-OH-Pyr (*m/z* 338.0) had the highest response, respectively (Figure S4). The responses of parent ions and daughter ions of pyriproxyfen and its five metabolites to different MS/MS collision energies are shown in Figure S5. The optimized MS/MS conditions of pyriproxyfen and its five metabolites are listed in Table 1, and the MS/MS spectra are shown in Figure S6.

*3.2. Optimization of Sample Extraction and Purification*

3.2.1. Optimization of Sample Extraction

Samples were prepared according to the modified QuEChERS method, and studies in the literature indicated that tea samples soaked in $H_2O$ prior to extraction can significantly improve the extraction efficiency of pesticide residues [34,46]. Therefore, the recoveries of pyriproxyfen and its metabolites were compared for different soaking solvents, i.e., $H_2O$ with 1%, 2% and 5% FA; with 1%, 2% and 5% Hac; and with 5% NaAc. The results showed that the recoveries of pyriproxyfen, PYPAC, DPH-Pyr, 5″-OH-Pyr and 4′-OH-Pyr in several different acidified water solvents were all greater than 70% with RSDs lower than 20%, while in $H_2O$ with 5% NaAc, PYPA could only obtain an appropriate recovery rate that met the analytical requirements, as shown in Figure 2A. Since there are four metabolites of pyriproxyfen, in order not to affect their extraction efficiency, in the subsequent salting-out step, we added NaAc to increase the recovery rate of PYPA in fresh tea leaves, tea and tea

infusion. Meanwhile, the effects of different volumes (0, 2.0, 5.0, 8.0 and 10.0 mL) of $H_2O$ with 5% FA on the recoveries of pyriproxyfen, PYPA, PYPAC, DPH-Pyr, 5″-OH-Pyr and 4′-OH-Pyr were compared and are shown in Figure S7A. It was found that the volume of $H_2O$ with 5% FA had certain effects on the recoveries of pyriproxyfen and its metabolites, and it could obtain the best recoveries when the volume was 10 mL. Due to the difference in the water contents of fresh tea leaves and tea (green tea and black tea), the impacts on the recoveries of these compounds were considered, and the final volumes of $H_2O$ with 5% FA of 5 mL and 10 mL were added to fresh tea leaves and tea, respectively.

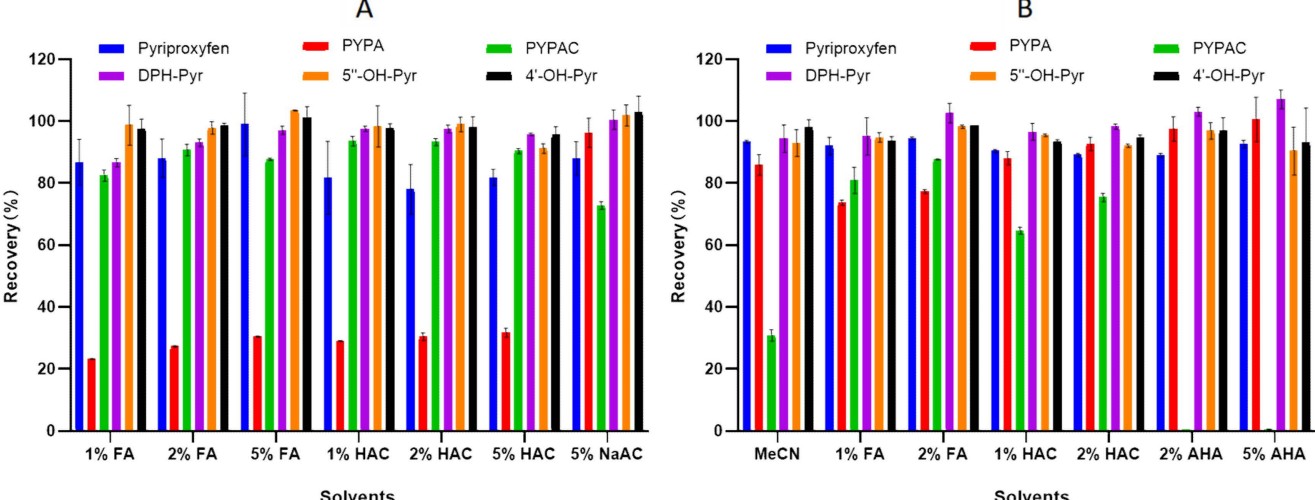

**Figure 2.** The recoveries of pyriproxyfen and its five metabolites in different soaking solvents (**A**) and different extraction solvents (**B**).

The properties of the target compounds, the polarities of the extraction solvents and the differences in sample matrices should be considered in the selection of the extraction solvent. MeCN is commonly used as an extraction solvent in the QuEChERS method, for it can dissolve organic, inorganic, gas and many other substances [47]. Due to the differences in the structures of pyriproxyfen and its metabolites, different pH values have a greater impact on the extraction effect. Therefore, MeCN was considered as the extraction solvent to compare the extraction recoveries of pyriproxyfen and its metabolites in black tea under different acidification conditions, i.e., MeCN with 1% FA, 2% FA, 1% HAc, 2% HAc, 2% $NH_3 \cdot H_2O$ and 5% $NH_3 \cdot H_2O$. The results in Figure 2B showed that the recoveries of pyriproxyfen and its metabolites PYPA, DPH-Pyr, 5″-OH-Pyr and 4′-OH-Pyr under several acidified MeCN conditions could meet the analytical requirements; PYPAC met the analysis requirements in FA- and HAc-acidified MeCN and had the best recovery in MeCN with 2% FA but not in MeCN, 2% $NH_3 \cdot H_2O$ in MeCN and 5% $NH_3 \cdot H_2O$ in MeCN. Therefore, MeCN with 2% FA was finally determined as the extraction solvent of pyriproxyfen and its metabolites in tea.

After extraction, NaCl was used for salting-out and stratification; NaAc was used for adjusting pH and increasing the recovery rate of PYPA; and $MgSO_4$ was used for removing excess water during soaking. Therefore, this study compared the recoveries of pyriproxyfen and its five metabolites in black tea with different usages of $MgSO_4$ (0, 2.0, 5.0, 8.0 and 10.0 g). It was found that the recoveries of pyriproxyfen and its metabolites (PYPA, 5″-OH-Pyr, PYPAC, 4′-OH-Pyr and DPH-Pyr) could meet the residue-analysis requirements and that the best recoveries were found when the quantity of $MgSO_4$ was 5.0 g, as shown in Figure S7B. At the same time, water infiltration can improve the extraction efficiency of MeCN for pesticides in sample matrices, but too much water can cause the incomplete extraction of pesticides using MeCN [48]. Due to the different water contents of fresh tea leaves and tea, water was added in the extraction step for soaking and improving the

extraction efficiency. So, the final usages of $MgSO_4$ for fresh tea leaves and tea were 3.0 g and 5.0 g, respectively.

### 3.2.2. Optimization of Sample Purification

Tea is rich in many components (such as polyphenols, caffeine, pigments, etc.) that affect the purification results. The purification effects and recoveries of pyriproxyfen and its metabolites in a series of different adsorbents (BC, PWAX, $C_{18}$, SCX, $C_{18}$-N, PSA, GCB and $MgSO_4$) and different quantities (5 mg–150 mg, where $MgSO_4$ was 10 mg—300 mg) were compared, and the results are shown in Figure S8. For the results meeting the recovery requirements of pyriproxyfen and its metabolites, $C_{18}$ was greater than PWAX, PSA and $MgSO_4$ and far greater than $C_{18}$-N, SCX, BC and GCB. The effects of $C_{18}$ and PWAX on the recoveries of pyriproxyfen and its metabolites considering different usages were slightly smaller than those of other adsorbents. Meanwhile, GCB was required to purify the pigments in tea. Therefore, the final adsorbents for purification were chosen as $C_{18}$, PWAX, GCB, and $MgSO_4$ with usages of 50, 10, 10 and 50 mg, respectively.

### *3.3. Method Validation*

#### 3.3.1. Linearities, MEs and LODs

The standard curves of pyriproxyfen and its five metabolites in the solvent and different tea matrices had good linearities, and the correlation coefficients (*r*) were equal to or above 0.9957, as shown in Table 2.

The MEs in fresh tea leaves, green tea, black tea, green-tea infusion and black-tea infusion were $-85 \sim -63\%$, $-79 \sim -42\%$, $-89 \sim -41\%$, $-47 \sim -3\%$ and $-35 \sim 40\%$, respectively. In black-tea infusion, matrix enhancement effects were noticed on pyriproxyfen, 5″-OH-Pyr, PYPAC and 4′-OH-Pyr, but on PYPA and DPH-Pyr, matrix suppression effects were noticed. On the other hand, on pyriproxyfen and its metabolites, matrix suppression effects were noticed in the other four matrices. Overall, the matrix-effect degrees for pyriproxyfen and its five metabolites were tea infusion < fresh tea leaves < tea.

Based on the responses to the lowest concentration levels on the matrix standard curves, the LODs of pyriproxyfen and its five metabolites that were calculated at $S/N = 3$ in tea matrices were less than 0.002 mg/L.

**Table 2.** Linear equations, correlation coefficients (*r*), matrix effects (MEs), limits of detection (LODs), average recoveries (ARs), standard deviations (SDs), relative standard deviations (RSDs) and limits of quantification (LOQs) of pyriproxyfen and its metabolites in different tea matrices.

| Compound | Matrix | Linear Range (mg/L) | Regression Equation | r | ME % | LOD (mg/L) | AR ± SD (%, n = 5) | | | | RSD (%) | | | | LOQ [e] (mg/L, mg/kg) |
|---|---|---|---|---|---|---|---|---|---|---|---|---|---|---|---|
| | | | | | | | C$_1$ [a] | C$_2$ [b] | C$_3$ [c] | C$_4$ [d] | C$_1$ [a] | C$_2$ [b] | C$_3$ [c] | C$_4$ [d] | |
| Pyriproxyfen | Solvent | 0.005~2.50 | y = 24534667 x + 1527353.0 | 0.9957 | — | 0.001 | — | — | — | — | — | — | — | — | — |
| | Fresh tea leaves | | y = 3779598 x + 142744.1 | 0.9968 | −85 | 0.002 | 91.9 ± 2.2 | 101.1 ± 4.0 | 93.6 ± 2.7 | 82.4 ± 1.6 | 2.4 | 4.0 | 2.9 | 1.9 | 0.002 |
| | Green tea | | y = 14281359 x − 84518.9 | 0.9999 | −42 | 0.002 | 80.3 ± 0.2 | 81.8 ± 1.9 | 84.5 ± 3.2 | 102.9 ± 2.2 | 0.3 | 2.3 | 3.8 | 2.1 | 0.005 |
| | Black tea | | y = 14393139 x + 178022.2 | 0.9995 | −41 | 0.002 | 79.9 ± 1.7 | 78.0 ± 0.7 | 84.6 ± 0.7 | 94.0 ± 2.7 | 2.1 | 0.8 | 0.9 | 2.9 | 0.005 |
| | Green-tea infusion | | y = 17512970 x + 243090.8 | 0.9964 | −29 | 0.001 | 86.0 ± 7.2 | 89.2 ± 7.6 | 82.2 ± 6.5 | 81.0 ± 8.4 | 8.3 | 8.5 | 7.9 | 10.3 | 0.0002 |
| | Black-tea infusion | | y = 25594622 x + 264354.6 | 0.9977 | 4 | 0.001 | 84.8 ± 8.1 | 81.8 ± 7.8 | 93.4 ± 9.3 | 78.0 ± 5.6 | 9.5 | 9.6 | 9.9 | 7.2 | 0.0002 |
| PYPA | Solvent | 0.005~2.50 | y = 8403221 x + 304731.1 | 0.9984 | — | 0.001 | — | — | — | — | — | — | — | — | — |
| | Fresh tea leaves | | y = 2495325 x + 32180.9 | 0.9992 | −70 | 0.002 | 90.0 ± 0.8 | 84.7±2.1 | 87.0 ± 1.4 | 82.9 ± 1.2 | 0.9 | 2.5 | 1.6 | 1.5 | 0.002 |
| | Green tea | | y = 1802508 x − 50455.2 | 0.9994 | −79 | 0.002 | 79.1 ± 2.2 | 77.0 ± 0.7 | 73.9 ± 3.5 | 83.7 ± 3.3 | 2.8 | 0.9 | 4.7 | 4.0 | 0.005 |
| | Black tea | | y = 947477 x + 71446.9 | 0.9996 | −89 | 0.002 | 71.8 ± 0.9 | 74.7 ± 1.1 | 74.1 ± 2.3 | 76.2 ± 2.7 | 1.3 | 1.5 | 3.2 | 3.5 | 0.005 |
| | Green-tea infusion | | y = 4432367 x + 13855.0 | 0.9994 | −47 | 0.002 | 86.0 ± 5.1 | 91.1 ± 6.1 | 89.8 ± 4.5 | 79.8 ± 7.0 | 6.0 | 6.6 | 5.0 | 8.8 | 0.0002 |
| | Black-tea infusion | | y = 5444457 x + 5498.9 | 0.9999 | −35 | 0.002 | 85.8 ± 11.3 | 91.7 ± 11.0 | 94.4 ± 10.5 | 81.0 ± 8.1 | 13.2 | 12.0 | 11.1 | 10.0 | 0.0002 |
| PYPAC | Solvent | 0.005~2.50 | y = 3443175 x + 40278.2 | 0.9998 | — | 0.001 | — | — | — | — | — | — | — | — | — |
| | Fresh tea leaves | | y = 1287068 x + 18715.5 | 0.9992 | −63 | 0.002 | 93.4 ± 2.6 | 86.4 ± 7.3 | 90.6 ± 4.7 | 85.1 ± 3.4 | 2.7 | 8.4 | 5.2 | 4.0 | 0.002 |
| | Green tea | | y = 1386492 x − 1294.9 | 1.0000 | −60 | 0.002 | 84.5 ± 3.7 | 85.2 ± 1.6 | 75.5 ± 4.8 | 73.0 ± 7.7 | 4.3 | 1.9 | 6.3 | 10.6 | 0.005 |
| | Black tea | | y = 881381 x + 29306.9 | 0.9993 | −74 | 0.002 | 78.9 ± 1.1 | 83.9 ± 1.8 | 97.1 ± 4.8 | 83.8 ± 7.4 | 1.4 | 2.2 | 4.9 | 8.9 | 0.005 |
| | Green-tea infusion | | y = 2147639 x + 10116.4 | 0.9992 | −38 | 0.002 | 99.1 ± 6.6 | 94.8 ± 6.0 | 91.9 ± 9.1 | 78.0 ± 7.7 | 6.6 | 6.3 | 9.9 | 9.9 | 0.0002 |
| | Black-tea infusion | | y = 3992739 x + 8829.5 | 0.9999 | 16 | 0.001 | 85.0 ± 7.6 | 90.5 ± 11.7 | 92.1 ± 8.5 | 74.7 ± 4.3 | 9.0 | 12.9 | 9.2 | 5.7 | 0.0002 |
| DPH-Pyr | Solvent | 0.005~2.50 | y = 10157350 x + 659675.9 | 0.9958 | — | 0.001 | — | — | — | — | — | — | — | — | — |
| | Fresh tea leaves | | y = 2443302 x − 969.9 | 0.9998 | −76 | 0.002 | 92.7 ± 2.2 | 89.9 ± 5.3 | 77.6 ± 2.2 | 81.7 ± 1.1 | 2.4 | 5.9 | 2.8 | 1.4 | 0.002 |
| | Green tea | | y = 5685209 x + 53600.9 | 0.9991 | −44 | 0.002 | 84.3 ± 3.7 | 87.1 ± 6.9 | 88.5 ± 4.4 | 90.1 ± 3.4 | 4.4 | 7.9 | 5.0 | 3.8 | 0.005 |
| | Black tea | | y = 3766814 x + 67246.9 | 0.9998 | −63 | 0.002 | 81.9 ± 4.8 | 84.4 ± 2.7 | 83.1 ± 2.1 | 89.1 ± 6.3 | 5.8 | 3.2 | 2.5 | 7.0 | 0.005 |
| | Green-tea infusion | | y = 6927646 x + 26399.5 | 0.9996 | −32 | 0.002 | 76.2 ± 2.2 | 77.1 ± 4.3 | 72.5 ± 6.9 | 71.2 ± 8.3 | 2.9 | 5.6 | 9.5 | 11.6 | 0.0002 |
| | Black-tea infusion | | y = 9651707 x + 15191.1 | 0.9995 | −5 | 0.001 | 79.4 ± 9.3 | 80.8 ± 6.1 | 85.5 ± 10.3 | 76.1 ± 11.0 | 11.7 | 7.5 | 12.1 | 14.4 | 0.0002 |

**Table 2.** *Cont.*

| Compound | Matrix | Linear Range (mg/L) | Regression Equation | r | ME % | LOD (mg/L) | AR ± SD (%, n = 5) | | | | RSD (%) | | | | LOQ [e] (mg/L, mg/kg) |
|---|---|---|---|---|---|---|---|---|---|---|---|---|---|---|---|
| | | | | | | | $C_1$ [a] | $C_2$ [b] | $C_3$ [c] | $C_4$ [d] | $C_1$ [a] | $C_2$ [b] | $C_3$ [c] | $C_4$ [d] | |
| 5″-OH-Pyr | Solvent | 0.005~2.50 | y = 17913728 x + 352196.5 | 0.9995 | — | 0.001 | — | — | — | — | — | — | — | — | — |
| | Fresh tea leaves | | y = 4356511 x + 51916.9 | 0.9994 | −76 | 0.002 | 92.5 ± 2.4 | 80.6 ± 4.2 | 81.0 ± 1.6 | 79.4 ± 2.7 | 2.6 | 5.2 | 1.9 | 3.4 | 0.002 |
| | Green tea | | y = 9449864 x − 97425.5 | 0.9999 | −47 | 0.002 | 82.9 ± 1.4 | 84.3 ± 0.9 | 82.0 ± 1.4 | 88.5 ± 3.7 | 1.7 | 1.1 | 1.7 | 4.2 | 0.005 |
| | Black tea | | y = 10043111 x + 117534.5 | 0.9998 | −44 | 0.002 | 75.3 ± 0.9 | 80.2 ± 1.7 | 78.4 ± 3.2 | 73.2 ± 1.6 | 1.2 | 2.2 | 4.1 | 2.2 | 0.005 |
| | Green-tea infusion | | y = 17407764 x + 98624.3 | 0.9985 | −3 | 0.001 | 86.2 ± 9.8 | 91.9 ± 6.7 | 79.6 ± 6.9 | 71.3 ± 3.7 | 11.3 | 7.3 | 8.7 | 5.3 | 0.0002 |
| | Black-tea infusion | | y = 25075650 x + 80662.0 | 0.9997 | 40 | 0.001 | 88.4 ± 3.3 | 78.4 ± 8.9 | 95.7 ± 3.4 | 94.3 ± 6.8 | 3.7 | 11.3 | 3.6 | 7.2 | 0.0002 |
| 4′-OH-Pyr | Solvent | 0.005~2.50 | y = 13673483 x + 777765.6 | 0.9970 | — | 0.001 | — | — | — | — | — | — | — | — | — |
| | Fresh tea leaves | | y = 3172569 x + 29566.8 | 0.9995 | −77 | 0.001 | 95.6 ± 2.4 | 85.0 ± 2.6 | 89.9 ± 2.2 | 83.7 ± 2.1 | 2.5 | 3.0 | 2.5 | 2.5 | 0.002 |
| | Green tea | | y = 7035067 x − 78002.7 | 0.9996 | −49 | 0.001 | 84.5 ± 1.2 | 86.2 ± 1.1 | 89.4 ± 3.8 | 94.2 ± 2.8 | 1.5 | 1.3 | 4.3 | 2.9 | 0.005 |
| | Black tea | | y = 6890730 x + 78033.0 | 0.9998 | −50 | 0.001 | 78.2 ± 0.7 | 81.2 ± 1.6 | 82.7 ± 4.1 | 86.9 ± 4.5 | 0.9 | 1.9 | 4.9 | 5.2 | 0.005 |
| | Green-tea infusion | | y = 11117912 x + 140268.6 | 0.9957 | −19 | 0.001 | 85.1 ± 7.5 | 86.5 ± 6.3 | 82.9 ± 8.3 | 76.1 ± 8.1 | 8.9 | 7.2 | 10.0 | 10.7 | 0.0002 |
| | Black-tea infusion | | y = 14847524 x + 180842.4 | 0.9969 | 9 | 0.001 | 81.4 ± 8.6 | 77.6 ± 3.4 | 88.3 ± 8.4 | 75.1 ± 5.0 | 10.5 | 4.4 | 9.5 | 6.6 | 0.0002 |

[a]: The spiked concentrations of pyriproxyfen and its five metabolites were 0.002 mg/kg, 0.005 mg/kg and 0.0002 mg/L in fresh tea leaves, green tea, black tea, green-tea infusion and black-tea infusion, respectively. [b]: The spiked concentrations of pyriproxyfen and its five metabolites were 0.01 mg/kg, 0.025 mg/kg and 0.001 mg/L in fresh tea leaves, green tea, black tea, green-tea infusion and black-tea infusion, respectively. [c]: The spiked concentrations of pyriproxyfen and its five metabolites were 0.1 mg/kg, 0.25 mg/kg and 0.01 mg/L in fresh tea leaves, green tea, black tea, green-tea infusion and black-tea infusion, respectively. [d]: The spiked concentrations of pyriproxyfen and its five metabolites were 1.0 mg/kg, 2.5 mg/kg and 0.10 mg/L in fresh tea leaves, green tea, black tea, green-tea infusion and black-tea infusion, respectively. [e]: The LOQs of pyriproxyfen and its five metabolites were 0.002 mg/kg, 0.005 mg/kg and 0.0002 mg/L in fresh tea leaves, green tea, black tea, green-tea infusion and black-tea infusion, respectively.

### 3.3.2. Recoveries, RSDs and LOQs

The results in Table 2 showed that when spiked at 0.002, 0.01, 0.1 and 1.0 mg/kg, in fresh tea leaves, the average recoveries of pyriproxyfen, DPH-Pyr, PYPA, 4′-OH-Pyr PYPAC and 5″-OH-Pyr were 77.6~101.1% with RSDs of 0.9~8.4%; when spiked at 0.005, 0.025, 0.250 and 2.50 mg/kg, the average recoveries in green tea were 73.0~102.9% with RSDs of 0.3~10.6%, and in black tea, they were 71.8~98.8% with RSDs of 0.8~8.9%. When spiked at 0.0002, 0.001, 0.010 and 0.10 mg/L, the average recoveries in green-tea infusion were 71.2~99.1% with RSDs of 2.9~11.6%, and in black-tea infusion, they were 74.7~95.7% with RSDs of 3.7~14.4%. The representative chromatograms of pyriproxyfen and its five metabolites in blank green tea, matrix standard and 0.025 mg/kg-spiked samples are shown in Figure 3.

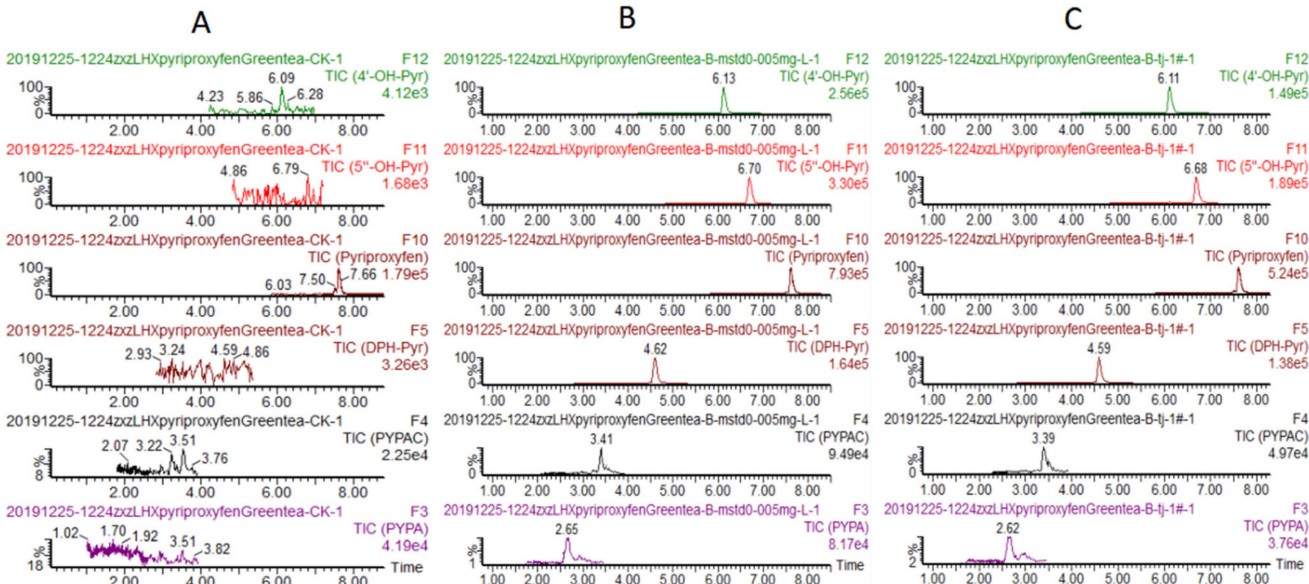

**Figure 3.** The chromatograms of pyriproxyfen and its five metabolites in blank green tea (**A**), matrix standard at 0.005 mg/L (**B**) and the 0.025 mg/kg-spiked level (**C**).

Defined as the lowest spike concentrations with suitable recoveries and RSDs, the LOQ of pyriproxyfen and its metabolites in fresh tea leaves was 0.002 mg/kg; in green tea and black tea, it was 0.005 mg/kg; in green-tea infusion and black-tea infusion, it was 0.0002 mg/L, respectively. As shown in Table 3, the LOQs found in this study were better than those found with many previously reported methods and far lower than the MRL of 15 mg/kg of pyriproxyfen in tea in the EU, the USA and Japan [49–51].

### 3.4. Application to Real Fresh Tea Leaf Samples from Field Test

The proposed method was performed to analyze the residues of pyriproxyfen and its five metabolites on fresh tea leaves after the application of 100 g/L pyriproxyfen EC to assess the applicability. Residues are shown in Table S2, and the dissipation curves of pyriproxyfen and its five metabolites on fresh tea leaves are shown in Figure 4.

The initial deposition of pyriproxyfen on fresh tea leaves was 14.55 mg/kg. Three days after application, the dissipation rate reached 86.7%, and the dissipation kinetic equation was $C = 8.7121\mathrm{e}^{-0.280t}$ with $R^2$ of 0.9587; the half-life $t_{1/2}$ was 2.48 d; the half-life of pyriproxyfen on fresh tea leaves was slightly less than that (2.74 d) of 30% pyriproxyfen chlorfenapyr suspension concentrate [52]. During the growth of fresh tea leaves, pyriproxyfen degraded and generated metabolites 4′-OH-Pyr, 5″-OH-Pyr, PYPA, DPH-Pyr and PYPAC. At different intervals after application, the residues of these five metabolites on fresh tea leaves were 0.007~0.054, 0.002~0.023, 0.003~0.035, 0.003~0.091 and 0.005~0.51 mg/kg, respectively. With the increase in the time interval since application, the residues of five metabolites on

fresh tea leaves first increased and then decreased; so, the dissipation of the metabolites did not conform to the first-order kinetic model, and the equation curve was not given.

**Table 3.** Comparison of existing analytical methods in terms of pyriproxyfen residues on different agricultural products.

| Agro-Products | Extraction Solvent | Detection Method | Mobile Phase | Column | LODs (mg/kg) | LOQs (mg/kg) | Reference |
|---|---|---|---|---|---|---|---|
| Tomatoes, green peppers | MeCN/$H_2O$ (80:20, *v/v*) | GC-NPD/MSD | / | HP-5MS (30 m × 0.25 mm, 0.25 μm) | / | / | [7] |
| Vegetables, fruits | Acidified MeCN | LC-MS/MS | 0.1% FA–MeCN, 2 mmol/L AMA and 0.1% FA–$H_2O$ | CAPCELLPAK $C_{18}$ (50 mm × 2.0 mm, 3 μm) | / | 0.005 | [19] |
| Oranges, strawberries, cherries, peaches, apricots, pears | Ethyl acetate | LC–MS/QqQ, LC–MS/QIT, LC–MS/QqTOF | 10 mm ammonium formate both in MeOH and $H_2O$ | Luna $C_{18}$ (150 mm × 4.6 mm, 5 μm) | / | 0.002 (QqQ), 0.02 (QIT), 0.05 (QqToF) | [20] |
| Fresh and canned peaches | Acetone-dichloromethane | LC-DAD | Acetonitrile/$H_2O$ (70:30, *v/v*) | $C_8$ Zorbax XDB Eclipse (4.6 mm × 150 mm, 5 μm) | / | 0.05 | [21] |
| Grapes | MeCN | HPLC-MS | 5 mm ammonium formate in $H_2O$ and 5 mm ammonium formate in MeOH | Zorbax SB $C_{18}$ (2.1 mm × 50 mm, 3.5 μm) | / | 0.05 | [22] |
| Peppers | Acetone, Ethyl acetate/cyclohexane (1/1, *v/v*) | GC-NPD | / | HP-5MSI (30 m × 0.25 mm, 0.25 μm) | / | / | [24] |
| Mushrooms | MeCN | UPLC-MS/MS | MeCN, 0.2% FA–$H_2O$ | BEH $C_{18}$ (2.1 mm × 50 mm, 1.7 μm) | 0.00016–0.00012 | 0.000052–0.00038 | [25] |
| Tomatoes | Ethyl acetate | HPLC-UV, Fluorometric | MeOH and MeCN (80:20, *v/v*), 272 nm | $C_{18}$ (4.6 mm × 250 mm, 5 μm) | 0.217 mg/L (UV) 0.146 mg/L (fluorometric) | 0.651 mg/L (HPLC-UV), 0.438 mg/L (fluorometric) | [26] |
| Tomatoes | MeCN | GC-MS/MS | / | HP-5 ms UI (30 m × 0.25 mm, 0.25 μm) | $2 \times 10^{-12}$ g | 0.01 | [27] |
| Citrus | MeCN | LC-MS/MS | MeOH, $H_2O$ | Poroshell 120 EC-$C_{18}$ (100 mm × 3 mm, 2.7 μm) | 0.01–0.02 | 0.005 | [28] |
| Fresh tea leaves, green tea, black tea, green-tea infusion and black-tea infusion | MeCN with 2% FA and $H_2O$ with 5% FA (QuEChERS method) | UPLC-MS/MS | MeOH with 0.1% FA and $H_2O$ with 0.1% FA | Acquity UPLC HSS T3 (100 mm × 2.1 mm, 1.8 μm) | 0.002 mg/L | 0.002 mg/kg in fresh tea leaves, 0.005 mg/kg in black tea and green tea, 0.0002 mg/L in black-tea infusion and green-tea infusion | Our method |

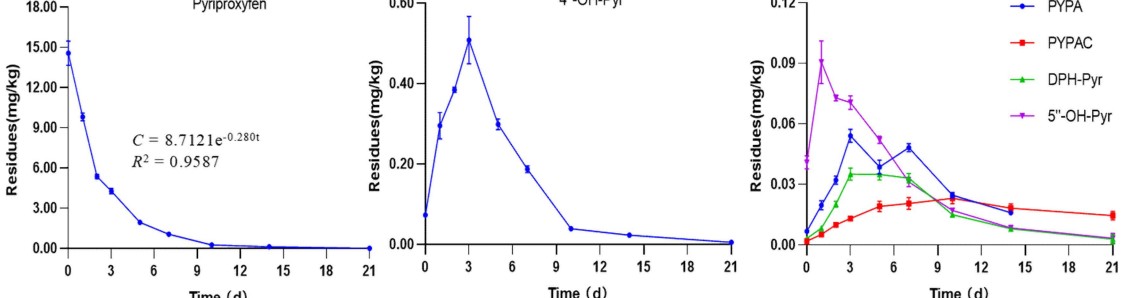

**Figure 4.** The dissipation trends of pyriproxyfen and its five metabolites on fresh tea leaves after application of 100 g/L pyriproxyfen EC.

## 4. Conclusions

A simultaneous residue-determination method for pyriproxyfen and its metabolites in fresh tea leaves, green tea, black tea, green-tea infusion and black-tea infusion was

established with the modified QuEChERs method for sample extraction and purification coupled with UPLC-MS/MS matrix external standard determination. Good linearities were obtained for pyriproxyfen and its five metabolites in different tea matrices with correlation coefficients (*r*) equal to or above 0.9957, and the LODs were less than 0.002 mg/L. The average spiked recoveries were 71.2~102.9% with RSDs of 0.3~14.4%; the LOQs were 0.002 mg/kg in fresh tea leaves, 0.005 mg/kg in green tea and black tea and 0.0002 mg/L in green-tea infusion and black-tea infusion, respectively. This proposed method was applied to real fresh tea leaves; the half-life of pyriproxyfen on fresh tea leaves was 2.48 d, and metabolites PYPA, PYPAC, DPH-Pyr, 5″-OH-Pyr and 4′-OH-Pyr were generated during the growth of fresh tea leaves in the field test. This established method was feasible and reliable, and it could meet the residue requirements for the MRLs of pyriproxyfen and its five metabolites in fresh tea leaves, tea and tea infusion.

**Supplementary Materials:** The following supporting information can be downloaded at: https://www.mdpi.com/article/10.3390/agronomy12081829/s1, Table S1: LC-MS and NMR information for metabolites PYPA, PYPAC, DPH-Pyr, 4′-OH-Pyr and 5″-OH-Pyr; Table S2: Residue dissipation of pyriproxyfen and its metabolites on fresh tea leaves; Figure S1: LC-MS and NMR spectrums of metabolites PYPA (a), PYPAC (b), DPH-Pyr (c), 4′-OH-Pyr (d) and 5″-OH-Pyr (e); Figure S2: Chromatogram of pyriproxyfen and its metabolites under the optimized chromatographic conditions; Figure S3: Responses of pyriproxyfen and its metabolites to different volume proportions of MeCN/H2O as the injection solvents; Figure S4: Responses of pyriproxyfen and its metabolites to different cone voltages; Figure S5: Responses of parent and daughter ions of pyriproxyfen and its metabolites to different collision energies; Figure S6: Fragmentation mass spectrometry of pyriproxyfen and its metabolites; Figure S7: Recoveries of pyriproxyfen and its metabolites in different volumes of $H_2O$ with 5% FA (A) and different quantities of $MgSO_4$ (B); Figure S8: Recoveries of pyriproxyfen and its five metabolites with different types and different usages of purification adsorbents.

**Author Contributions:** All authors contributed to the study conception and design; experiment, software, data curation and analysis, visualization and writing—original draft—were performed by H.L.; field test, sample processing, investigation and formal analyses were performed by Q.Z., X.W., F.L. and L.Z.; supervision and advising were performed by Z.C. X.Z. is the corresponding author and performed project administration, methodology, conceptualization, validation, formal analysis and writing—review and editing. All authors commented on previous versions of the manuscript. All authors have read and agreed to the published version of the manuscript.

**Funding:** This work was financially supported by National Natural Science Foundation of China (No. 31772077) and the Innovation Program of Chinese Academy of Agricultural Sciences (CAAS-ASTIP-2016-TRI).

**Institutional Review Board Statement:** Not applicable.

**Informed Consent Statement:** Not applicable.

**Data Availability Statement:** Not applicable.

**Acknowledgments:** Not applicable.

**Conflicts of Interest:** The authors declare no conflict of interest.

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
