# Peer review of "Simultaneous Quantitative Determination of Residues of Pyriproxyfen and Its Metabolites in Tea and Tea Infusion Using Ultra-Performance Liquid Chromatography–Tandem Mass Spectrometry"

_agronomy, doi:10.3390/agronomy12081829_

Round 1

Reviewer 1 Report

The publication has serious errors and shortcomings and cannot be accepted for publication in this form. Moreover, the authors have already published an analogous paper this year https://doi.org/10.1002/jsfa.11746! I do not understand why they are duplicating their results. In lines 80-88 the authors write that their aim is to develop a NEW method, while this method is already published by them. So what is the novelty of the presented research?

lines 57-67: please refer here to the fact that this compound has already been determined in tea and by the same authors, citation 52

How were the product ions selected, where are the fragmentation spectra?

Figures 2 and 3 are not clear, please transfer part of the data to the supplement

lines 327-333: methods with so great suppression by matrix effect are not appropriate, not useful, and should be refined. The authors cannot talk/write about the success of their research until they have refined the method of purification so that the matrix effect is much smaller.

Figure 4, please provide a TIC chromatogram.

Conclusions: this sounds like a summary of the paper and not the conclusions of the study, please revise.

Author Response

The publication has serious errors and shortcomings and cannot be accepted for publication in this form. Moreover, the authors have already published an analogous paper this year https://doi.org/10.1002/jsfa.11746! I do not understand why they are duplicating their results. In lines 80-88 the authors write that their aim is to develop a NEW method, while this method is already published by them. So what is the novelty of the presented research?

Response: The emphasis of this manuscript is different from that of the published article (https://doi.org/10.1002/jsfa.11746). This manuscript focuses on the establishment of analytical method. The published article focused on the residue degradation study of pyriproxyfen in the process of different tea with this method, which does not involve a complete methodological study. In fact, the submission of this manuscript has always been in the front, but due to various reasons, it has fallen behind the published article after continuous resubmission.

lines 57-67: please refer here to the fact that this compound has already been determined in tea and by the same authors, citation 52.

Response: Thank you for your suggestion. We have also mentioned this in the manuscript.

How were the product ions selected, where are the fragmentation spectra?

Response: Thank you for your wonderful question. The results have been given out in Suppl. Figure S5 and Suppl. Figure S6.

Figures 2 and 3 are not clear, please transfer part of the data to the supplement.

Response: Thank you for your wonderful question. In the revised manuscript, Figure 3 has been transferred to the supplement.

lines 327-333: methods with so great suppression by matrix effect are not appropriate, not useful, and should be refined. The authors cannot talk/write about the success of their research until they have refined the method of purification so that the matrix effect is much smaller.

Response: Thank you for your good question. In fact, in the pesticide and their metabolites residue analysis, it is well known that tea is a very complex matrix, and it is difficult to completely remove the matrix effect. Therefore, the general method is to remove the matrix effect as much as possible, and use the matrix standard for quantitative analysis.

Figure 4, please provide a TIC chromatogram.

Response: In the tandem mass spectrometry method, TIC chromatogram is unnecessary and meaningless, so a TIC chromatogram was not provided.

Conclusions: this sounds like a summary of the paper and not the conclusions of the study, please revise.

Response: Thank you for your wonderful suggestion. In the revised manuscript, the part of conclusions has been modified.

Reviewer 2 Report

The manuscript by Li et. al, is well documented and I recommend its publication after some minor revisions on the following points:

1) Introduction Line 57: Write "chromatrographic analytical methods combined with various spectroscopic techniques"

2) Line 325: the correlation coefficient is symbolized by "r". If the authors actualy provide in the manuscript the R2 value, then these values should be substituted by r values

3) Table 2: In the regression equations the authors should provide the standard deviations for the slope and the intercept. Additionally, in the supplemental material should present the calibration curves.

4) Figure 4(A): Concerning the TIC for DPH-Pyr, the peak at 4,59 min is in the limits of noise, while for PYPA the peak is not visible at all. The authors should provide better TIC's and the MS data for these two compounds in black green tea otherwise we cannot be sure for the correct detection of these compounds.

Author Response

Comments and Suggestions for Authors

The manuscript by Li et. al, is well documented and I recommend its publication after some minor revisions on the following points:

1) Introduction Line 57: Write "chromatographic analytical methods combined with various spectroscopic techniques"

Response: Thank you very much. In the revised manuscript, this sentence has been changed.

2) Line 325: the correlation coefficient is symbolized by "r". If the authors actually provide in the manuscript the R2 value, then these values should be substituted by r values

Response: Thank you very much. In the revised manuscript, all of these have been changed.

3) Table 2: In the regression equations the authors should provide the standard deviations for the slope and the intercept. Additionally, in the supplemental material should present the calibration curves.

Response: Thank you very much for your question. In fact, in this manuscript, the average value of each concentration point was used to get the curve equation. The regression curve equations were given in Table 2, including the slope and the intercept. Therefore, the standard calibration curves were not given in the supplemental materials.

4) Figure 4(A): Concerning the TIC for DPH-Pyr, the peak at 4.59 min is in the limits of noise, while for PYPA the peak is not visible at all. The authors should provide better TIC's and the MS data for these two compounds in black green tea otherwise we cannot be sure for the correct detection of these compounds.

Response: Thank you very much for your good question. Yes, for PYPA, there was a high background baseline signal in the blank green tea sample, but it was found that there was an obvious chromatographic peak in the matrix standard sample or the spiked sample, the baseline signal was not high. At the same time, in order to ensure that the spectra of pyriproxyfen and its metabolites come from the same blank green tea sample, so, the corresponding spectra of PYPA was not changed.

Reviewer 3 Report

Interesting paper regarding the possible contamination of tea, a highly consumed drink around the world. A suggestion is that the method should be applied to other types of teas as a comparable study.

-          Green tea and black tea have the same LOQ and LODs? Where calculated separately or in the same assay and formula?

-          Please provide the names of the metabolites in the first time they are referred in the text and before presenting only the abbreviations (4'-OH-Pyr, 2-OH-PY, DPH-Pyr, and 4''-OH-POP).

-          Line 30: “the maximum residue limits (MRLs) of pesticides on tea have changed at home and abroad”

-          What does that mean by home and abroad?

-          Line 63 – 67: Maybe add some considerations regarding the limits of detection of those different methods

-          Line 117 – “Then stored at -20°C until use.” For how long? Any stability studies?

-          Line 181 – Authors should explain how those parameters were evaluated: sensitivities, recoveries, precisions, and specificities (as it is

-          Line 241 – Do not begin a sentence with “and” (“And pyriproxyfen…”)

Author Response

Comments and Suggestions for Authors

Interesting paper regarding the possible contamination of tea, a highly consumed drink around the world. A suggestion is that the method should be applied to other types of teas as a comparable study.

Response: Thank you for your wonderful suggestion. According to the processing, there are six categories of different tea. In the research, green tea (non fermented tea) and black tea (fully fermented tea) with the largest output and consumption were taken as the representative samples for analysis, which can also basically represent all tea samples.

-Green tea and black tea have the same LOQ and LODs? Where calculated separately or in the same assay and formula?

Response: Thank you for your wonderful question. In fact, the LOQs of this method were defined as the actual minimum spiked concentration in green tea and black tea. Therefore, in order to consider the operability of this method, the same spiked concentration was achieved, so the LOQs in green tea and black tea were same in this manuscript.

-Please provide the names of the metabolites in the first time they are referred in the text and before presenting only the abbreviations (4'-OH-Pyr, 2-OH-PY, DPH-Pyr, and 4''-OH-POP).

Response: Thank you very much for your wonderful suggestion. In the part of “2.1. Materials and reagents”, all of the ACS No. and chemical formulas were given out. It was known out that the CAS registration number of compound was unique.

-Line 30: “the maximum residue limits (MRLs) of pesticides on tea have changed at home and abroad”. What does that mean by home and abroad?

Response: Thank you very much for your wonderful suggestion. In the revised manuscript, this sentence has been changed and modified.

-Line 63 – 67: Maybe add some considerations regarding the limits of detection of those different methods.

Response: Thank you very much for your wonderful suggestion. The limits of detection and limits of quantification of those different methods have been given out in Table 3.

-Line 117 – “Then stored at -20°C until use.” For how long? Any stability studies?

Response: Thank you very much for your good question. In fact, the storage time was not exceed 3 months. The stability study of pyriproxyfen has been carried out, and the results showed that the storage of pyriproxyfen was not changed significantly for a year.

-Line 181 – Authors should explain how those parameters were evaluated: sensitivities, recoveries, precisions, and specificities (as it is

Response: Thank you very much for your good question. In the part of “3.3. Method validation”, all of those parameters have been evaluated.

-Line 241 – Do not begin a sentence with “and” (“And pyriproxyfen…”)

Response: Thank you very much for your wonderful suggestion. In the revised manuscript, this sentence has been changed and modified.